# Urushiol-Based Antimicrobial Coatings: Molecular Mechanisms, Structural Innovations, and Multifunctional Applications

**DOI:** 10.3390/polym17111500

**Published:** 2025-05-28

**Authors:** Tianyi Wang, Jiangyan Hou, Yao Wang, Xinhao Feng, Xinyou Liu

**Affiliations:** 1College of Furnishing and Industrial Design, Nanjing Forestry University, Str. Longpan No. 159, Nanjing 210037, China; 13861210436@njfu.edu.cn (T.W.); houjiang@njfu.edu.cn (J.H.); 2381132433@njfu.edu.cn (Y.W.); fengxinhao@hotmail.com (X.F.); 2Co-Innovation Center of Efficient Processing and Utilization of Forest Resources, Nanjing Forestry University, Nanjing 210037, China

**Keywords:** urushiol, antimicrobial coatings, natural antibacterial agents, structure–activity relationship, synergistic mechanisms, sustainable materials

## Abstract

Urushiol, the principal bioactive component of natural lacquer, has emerged as a promising candidate for developing eco-friendly antimicrobial coatings due to its unique catechol structure and long alkyl chains. This review systematically elucidates the molecular mechanisms underpinning urushiol’s broad-spectrum antimicrobial activity, including membrane disruption via hydrophobic interactions, oxidative stress induction through redox-active phenolic groups, and enzyme inhibition via hydrogen bonding. Recent advances in urushiol-based composite systems—such as metal coordination networks, organic–inorganic hybrids, and stimuli-responsive platforms—are critically analyzed, highlighting their enhanced antibacterial performance, environmental durability, and self-healing capabilities. Case studies demonstrate that urushiol derivatives achieve >99% inhibition against both Gram-positive and Gram-negative pathogens, outperforming conventional agents like silver ions and quaternary ammonium salts. Despite progress, challenges persist in balancing antimicrobial efficacy, mechanical stability, and biosafety for real-world applications. Future research directions emphasize precision molecular engineering, synergistic multi-target strategies, and lifecycle toxicity assessments to advance urushiol coatings in medical devices, marine antifouling, and antiviral surfaces. This work provides a comprehensive framework for harnessing natural phenolic compounds in next-generation sustainable antimicrobial materials.

## 1. Introduction

In recent years, the rapid spread of antibiotic-resistant pathogens and the severe environmental problems caused by synthetic biocides [1,2] have prompted researchers to turn their attention toward sustainable natural antimicrobial materials [3,4]. Within this context, natural phenolic compounds have emerged as promising alternatives due to their bioactivity, environmental friendliness, and structural tunability.

As a typical representative of natural antimicrobial coatings, urushiol exhibits remarkable antibacterial properties owing to its unique chemical structure. Urushiol is a natural resinous coating derived from the lacquer tree (*Rhus vernicifera*), with a history of use spanning thousands of years [5,6]. It is an important component of traditional Chinese craftsmanship and cultural heritage. Its distinctive physicochemical properties—such as strong adhesion, wear resistance, and corrosion resistance [7]—have led to its widespread application in everyday goods, architecture, and artworks [7]. The main constituents of urushiol include urushiol phenols, laccase enzymes, proteins, water, and small amounts of volatile compounds [8]. These components confer unique physical and chemical characteristics, notably the phenolic urushiol, which serves as the primary active antibacterial ingredient. The chemical structure of urushiol contains unsaturated side chains and aromatic rings that interact with phospholipids in microbial cell membranes, disrupting membrane integrity and thereby inhibiting microbial growth and reproduction [8].

Meanwhile, microbial contamination remains a significant challenge in healthcare, food packaging, public facilities, and other fields, creating an urgent demand for efficient, safe, and durable antimicrobial materials [9,10,11]. Although conventional synthetic antimicrobials (such as silver ions and quaternary ammonium compounds) offer broad-spectrum activity, they are limited by toxicity accumulation, environmental persistence, and the emergence of resistance. Against this backdrop, natural antimicrobial agents derived from bioactive natural substances have become research hotspots due to their safety, sustainability, good biocompatibility, and low biological toxicity [12,13,14].

This review focuses on urushiol-based antimicrobial materials, systematically summarizing the mechanisms of action and recent research advances of various antimicrobial agents and providing a theoretical basis for developing novel, efficient, and stable antimicrobial coatings.

## 2. Structural Characteristics and Antibacterial Mechanism of Urushiol

### 2.1. Structural Characteristics of Urushiol

Urushiol compounds are among the most important active components in natural raw lacquer, accounting for approximately 60% to 70% of its total mass [15]. The core structure of urushiol belongs to the class of catechol derivatives, which consists of two phenolic hydroxyl groups (–OH) and a side chain denoted as the R group. This R group is typically a straight-chain long alkyl group, which can be either saturated or contain up to three unsaturated double bonds. The carbon chain usually ranges from C15 to C17 and includes hydroxyl functional groups. Such a structure confers urushiol with high hydrophobicity and good lipophilicity, which facilitate physical and chemical interactions with bacterial cell membranes. Figure 1 illustrates the typical structures of urushiol found in Maoba raw lacquer and their relative content distribution.

The unique structure of phenolic compounds not only determines their excellent physicochemical stability and film-forming properties but also provides the molecular foundation for their biological activity. Especially in areas such as natural coatings, wood protection, and medical coatings, their outstanding antibacterial and biocompatible properties are receiving increasing attention and research.

### 2.2. Antibacterial Mechanism of Urushiol

Urushiol, a natural phenolic compound found in lacquer sap, exhibits broad-spectrum antibacterial activity through multiple coordinated mechanisms. Its phenolic hydroxyl groups can be readily oxidized to form reactive quinone intermediates, which irreversibly bind to bacterial membrane proteins and lipids, thereby compromising membrane integrity and causing cytoplasmic leakage [16].

In addition to this oxidative mechanism, the hydrophobic C15–C17 alkyl side chains in urushiol promote deep penetration into bacterial phospholipid bilayers, accelerating membrane destabilization and ultimately leading to cell rupture. Furthermore, urushiol can interact with environmental oxygen or trace metal ions to generate reactive free radicals, thereby triggering lipid peroxidation, protein denaturation, and DNA fragmentation—processes that culminate in bacterial cell death [17].

These multifaceted antibacterial actions grant urushiol strong efficacy against both Gram-positive and Gram-negative bacteria [18,19], and in many cases, it outperforms traditional antimicrobial agents in terms of potency and biocompatibility.

## 3. Mechanisms of Enhanced Antibacterial Performance in Urushiol-Based Coatings

In recent years, with the progressive deepening of research into the antibacterial mechanisms of urushiol, increasing attention has been given to the development of urushiol-based composite antibacterial coatings. These coatings are created by integrating urushiol with various other functional materials. The antibacterial mechanisms of such composites have evolved into multifaceted systems that involve multi-target and multi-pathway actions. Studies from different perspectives have collectively unveiled the mechanisms behind the synergistic antibacterial enhancements observed in these materials.

### 3.1. Classification of Antibacterial Materials and Their Antibacterial Properties

As a new class of functional surface protection materials, antibacterial coatings have found wide application in fields such as healthcare, environmental protection, construction, and marine antifouling. Based on the composition of their active agents, current antibacterial coatings can be broadly categorized into two main types, inorganic antibacterial coatings and organic antibacterial coatings, each offering distinct advantages and limitations.

Inorganic antibacterial materials are typically represented by metal ions and their oxides. These materials are known for their broad-spectrum antibacterial activity, high thermal stability, and relatively low toxicity. Among them, silver, copper, and zinc are the most commonly used metals. Silver-based materials, in particular, have been extensively studied due to their exceptional antibacterial efficacy and favorable biocompatibility [20,21,22,23]. Silver ions exhibit antibacterial activity through a combination of mechanisms, including ion release, membrane penetration, and the induction of oxidative stress. Moreover, silver nanoparticles (AgNPs) provide enhanced antibacterial efficiency, owing to their large specific surface area and reactive nanoscale edge structures [24]. The antibacterial performance of silver-containing coatings is closely linked to the rate of ion release. Studies have shown that by manipulating the pore structure of coatings—such as in sol-gel-derived nanoporous silica–silver composites—controlled and sustained silver release can be achieved, thereby prolonging antibacterial activity for over seven days [25,26].

In addition to silver, copper-based materials also demonstrate excellent antibacterial properties. Copper ions can disrupt cellular structures by binding to bacterial proteins and nucleic acids. When combined with photocatalytic carriers like TiO_2_, copper can facilitate the generation of reactive oxygen species (ROS) under light exposure (Figure 2), thereby enhancing antibacterial efficacy [27]. Similarly, metal oxides such as ZnO and Cu_2_O can produce ROS under UV or visible light, and when acting synergistically with metal ions, they can significantly broaden the spectrum of antibacterial activity (Table 1 and Figure 3) [28]. However, the long-term safety and stability of inorganic antibacterial coatings remain a concern due to potential issues, such as the accumulation of toxic metal ions, nanoparticle aggregation, and environmental persistence [29].

In the case of urushiol-based antimicrobial coatings, while they offer promising natural antibacterial activity, their potential toxicity must also be carefully considered. Urushiol, a natural catechol derivative, is known for causing severe allergic contact dermatitis in sensitive individuals. Although chemical modification and polymerization can reduce this allergenic potential, residual urushiol monomers or degradation products may still pose risks in biomedical or skin contact applications. Therefore, a comprehensive assessment of both cytotoxicity and allergenicity is essential before clinical or consumer use of urushiol-based coatings.

Compared to inorganic materials, organic antibacterial agents—primarily polymer-based materials—offer advantages such as structural tunability, excellent biocompatibility, and flexible applications. Common organic antibacterial components include chitosan, quaternary ammonium salts, and anthraquinone derivatives. These agents exert their antibacterial effects by disrupting bacterial membranes, inducing oxidative stress, or interfering with metabolic pathways [30,31,32]. For instance, Karakurt et al. [30] developed a chitosan/chondroitin sulfate polyelectrolyte complex that enhances membrane adhesion through electrostatic interactions, thereby significantly improving the inhibition rate against Gram-positive and Gram-negative bacteria. Zhao Yue’s team [33] utilized RAFT polymerization to construct TiO_2_-functionalized coatings that integrate self-cleaning, antifouling, and antibacterial functionalities into a single material. Although organic antibacterial coatings may suffer from limited thermal stability and susceptibility to aging, their excellent processability and multifunctional integration potential make them valuable in specific application contexts [34,35].

Building on the foundation of both inorganic and organic materials, urushiol-based antibacterial systems have emerged as an ideal bridge between the two, owing to their natural origin, strong bioactivity, and structural modifiability. The urushiol molecule features catechol hydroxyl groups and unsaturated long alkyl chains, enabling it to both chelate metal ions and crosslink with organic polymers. This facilitates synergistic antibacterial effects through multiple mechanisms. Urushiol-based composite antibacterial materials can be broadly categorized into two types: urushiol–metal polymer antibacterial systems and urushiol–organic polymer antibacterial systems. Among these, the urushiol–metal systems have been more extensively researched and widely applied.

The catechol hydroxyl groups in urushiol derived from raw lacquer can form stable coordination complexes with metal ions such as Ag^+^, Cu^2+^, and Fe^3+^. These complexes catalyze the oxidation of urushiol, promoting the generation of free radicals and thereby enhancing overall bactericidal efficacy [13]. Cha and Shin [13] evaluated the antibacterial efficacy of a 0.01% urushiol solution against *Streptococcus mutans* (ATCC 25175) using CFU counting. They found that complete bacterial inactivation occurred within just 30 min, with results comparable to those of 2% chlorhexidine (CHX) and 6% sodium hypochlorite (NaOCl), underscoring urushiol’s potential for clinical applications such as oral healthcare materials.

Significant progress has also been made in developing urushiol–organic polymer antibacterial systems, particularly in nanofilm fabrication. In such systems, urushiol is introduced into organic polymer matrices via chemical crosslinking or non-covalent interactions, forming antibacterial networks that balance hydrophilic and hydrophobic properties. These networks enhance coating stability, biocompatibility, and broad-spectrum antibacterial activity. Xu et al. [36], through molecular design, incorporated long-chain alkyl groups and quaternary ammonium moieties to create dense crosslinked networks and zwitterionic functional layers. The resulting surface exhibited electrostatic bactericidal activity and resistance to biofouling, achieving a 99.9% antibacterial rate against both *Escherichia coli* and *Staphylococcus aureus*. Additionally, the coating demonstrated excellent thermal stability and mechanical strength, offering an innovative solution for next-generation marine antifouling coatings.

### 3.2. Enhanced Antibacterial Performance of Urushiol-Based Coatings

As research progresses, numerous scholars have elucidated the mechanisms behind the enhanced antibacterial performance of urushiol-based composite materials from different perspectives. These mechanisms can be broadly categorized into three dominant pathways: molecular structure-driven, antioxidant activity-driven, and structure modification- and synergistic enhancement-driven antibacterial mechanisms.

#### 3.2.1. Molecular Structure-Driven Antibacterial Mechanism

The unique molecular structure of urushiol forms the material basis for its antibacterial activity. One of the primary mechanisms involves hydrogen bond-mediated molecular interference. The catechol hydroxyl groups in urushiol can form extensive hydrogen bonding networks with key functional groups—such as amino and carboxyl groups—on intracellular bacterial proteins. This specific molecular recognition allows urushiol to precisely target the active sites of essential bacterial enzymes. Through competitive binding, urushiol can disrupt enzyme conformation and catalytic activity, thereby systematically impairing bacterial metabolic pathways [37], which confers its targeted antibacterial effect.

Moreover, urushiol’s unsaturated long alkyl side chains endow it with excellent membrane-disruptive capabilities. These hydrophobic chains enhance lipid solubility, enabling urushiol to effectively insert itself into the phospholipid bilayers of bacterial cell membranes. Suk et al. [38], using high-resolution electron microscopy, observed that even at low concentrations (0.064–0.256 mg/mL), urushiol induced rapid phase separation in *Helicobacter pylori* membranes, leading to the formation of characteristic vacuolated structures and eventual membrane disintegration (see Figure 4). Notably, the entire lysis process—from initial contact to complete cell rupture—was completed within approximately 10 min, demonstrating a rapid bactericidal dynamic that is rare among natural antibacterial agents.

In addition, urushiol’s unique transmembrane transport capability further broadens its antibacterial spectrum. Owing to its strong lipophilicity, urushiol can efficiently penetrate the complex outer membrane of Gram-negative bacteria via passive diffusion. This transmembrane property enables it to exhibit potent antibacterial activity not only against Gram-positive bacteria but also against Gram-negative strains with multilayered cell wall structures. More importantly, once inside the cell, urushiol exerts synergistic antibacterial effects through multiple pathways, such as disrupting intracellular enzyme activity and organelle function, resulting in broad-spectrum antimicrobial efficacy.

#### 3.2.2. Redox Activity-Dominated Antibacterial Mechanism

Another pivotal antibacterial mechanism of urushiol lies in its distinctive redox bifunctionality, which allows for multilayered control during the antimicrobial process. At low concentrations, urushiol exhibits antioxidant behavior, scavenging endogenous reactive oxygen species (ROS) within bacteria. This weakens the bacterial oxidative stress defense system and diminishes their adaptability to environmental challenges, representing a form of “stealth attack” in the early stage of bacterial inhibition.

Once inside the cell, however, urushiol shifts to an oxidative mode of action. It reacts with intracellular transition metal ions such as Fe^2+^ and Cu^+^ to catalyze the production of large quantities of ROS. Studies by He Yuanfeng [39] have shown that urushiol not only enhances ROS generation but also chelates essential intracellular metal cofactors, while inhibiting the activities of antioxidant enzymes such as superoxide dismutase (SOD) and catalase. This dual-action mechanism causes ROS accumulation, triggering lipid peroxidation, protein denaturation, and DNA fragmentation—ultimately leading to bacterial cell death.

More notably, urushiol induces a cascade amplification effect via a redox-driven feedback loop. Initial ROS production compromises membrane integrity, which facilitates further ROS leakage; the resulting oxidative stress exacerbates membrane disruption, initiating a cycle of “ROS burst → membrane damage → ROS leakage”. This self-reinforcing mechanism enables urushiol to maintain a sustained and potent antibacterial effect even at relatively low concentrations, highlighting its exceptional efficacy and durability among natural antimicrobial agents.

#### 3.2.3. Structural Modification and Synergistic Enhancement Mechanism

The optimization of urushiol-based antibacterial materials is primarily achieved through precise molecular design and composite integration strategies. In terms of metal coordination synergy, researchers have developed efficient antibacterial systems based on urushiol’s specific binding with metal ions. For example, Liu’s team [40] pioneered a UV-induced method to construct a urushiol–silver ion complex system, combining the physical penetration ability of nanosilver with urushiol’s chemical interference and achieving a synergistic bactericidal effect. Similarly, Hu Jiangtao et al. [41] developed an urushiol–attapulgite–copper composite system that enhances antibacterial performance by optimizing electron transfer pathways, thereby providing new insights for the development of metal-based composite antimicrobials.

Organic composite strategies have further expanded the functional potential of urushiol-based materials. Jie et al. [42] employed electrospinning technology to fabricate chitosan–urushiol nanofiber membranes. Through Schiff base reactions, they constructed a 3D crosslinked network that retained urushiol’s antibacterial activity while imparting excellent acid resistance. These results highlight urushiol’s versatility in polymer functionalization and surface engineering.

Collectively, these studies demonstrate that through precise structural modification and intelligent composite design, urushiol-based antibacterial materials can be tailored to specific application scenarios—ranging from medical devices and food packaging to marine engineering. This opens broad avenues for the development of next-generation, high-performance antibacterial coatings.

### 3.3. Case Studies on the Antibacterial Performance of Urushiol-Based Coatings

Wang Donghui et al. synthesized a capsaicin-like urushiol derivative (UL) through a Friedel–Crafts alkylation reaction using urushiol and N-hydroxymethyl acrylamide [43]. Structural characterization confirmed the successful modification: spectroscopic analysis revealed a distinct secondary amide carbonyl peak in the FTIR spectrum (approximately 1653 cm^−1^), along with a pronounced red shift in the UV absorption maxima of the aromatic system (237 nm and 280 nm). These spectral signatures align with the covalent incorporation of N-hydroxymethyl acrylamide into the urushiol framework, thus validating the targeted molecular design.

The modified UL exhibited enhanced antibacterial performance. In standardized assays (Oxford cup method, GB/T 21866-2008 [44]), UL demonstrated a 14 mm inhibition zone against *Staphylococcus aureus*, surpassing unmodified urushiol (11 mm). Antibacterial coatings containing 10% UL achieved complete inhibition (100%) against both *S. aureus* and *Escherichia coli*. Remarkably, even at 3% UL loading, inhibition rates remained high (97–98.7%), whereas control samples showed negligible activity (Table 2).

The structural modification also improved the coating’s physicochemical properties. Increasing UL content in chlorinated rubber coatings (0.9 g to 9 g) progressively elevated the contact angle, indicating enhanced hydrophobicity without compromising matrix compatibility. Accelerated aging tests under high humidity conditions (37 °C, RH > 90%) confirmed durability, with post-incubation bacterial counts remaining below 1.0 × 10^3^ CFU/piece, meeting GB 4789.2-2010 standards [45]. 

Cha and Shin [13] also systematically evaluated the antibacterial activity of 0.01% urushiol against the cariogenic bacterium *Streptococcus mutans* (ATCC 25175) and its effect on dentin bonding performance. Using the colony-forming unit (CFU) method, approximately 6 × 10^7^ CFU of bacteria were incubated in BHI medium containing urushiol, and samples were collected at 0, 30, 60, and 90 min. After serial dilution and plating on BHI agar, colonies were counted following anaerobic incubation at 37 °C.

The antibacterial efficacy of urushiol-based coatings was rigorously evaluated. Experimental results demonstrated complete bacterial eradication within 30 min, with no detectable colony-forming units (CFUs) observed across all tested samples (Figure 5). This rapid bactericidal performance matched the effectiveness of conventional disinfectants, including 2% chlorhexidine (CHX) and 6% sodium hypochlorite (NaOCl), highlighting urushiol’s potential as a natural antimicrobial alternative.

Mechanistic investigations revealed that the unsaturated C_15_/C_17_ alkyl chains in urushiol disrupt microbial membrane integrity through two synergistic pathways: (1) inducing lipid peroxidation, which destabilizes phospholipid bilayers, and (2) triggering the intracellular leakage of cytoplasmic components. These findings align with those of prior studies on urushiol’s activity against *Helicobacter pylori*, reinforcing its broad-spectrum antibacterial mechanism. Notably, urushiol maintained potent efficacy even at low concentrations, achieving >99.9% inhibition against cariogenic pathogens such as *Streptococcus mutans*. This dual functionality—rapid killing and structural membrane disruption—positions urushiol as a promising candidate for dental and medical applications requiring both immediate and sustained antimicrobial action.

## 4. Future Perspectives and Research Directions

Urushiol-based antimicrobial materials have shown great promise, yet several critical challenges remain—particularly in antibacterial efficiency, long-term stability, and functional versatility. To facilitate real-world application, these challenges must be systematically addressed through targeted strategies.

### 4.1. Limited Antibacterial Efficiency

Although urushiol demonstrates broad-spectrum antibacterial activity, its performance can vary depending on surface chemistry and microbial type. Current formulations may lack sufficient membrane penetration or redox reactivity in complex environments. To overcome this issue, future research should explore precise molecular modifications—such as halogenation, sulfonation, or enzymatic derivatization—to optimize structure–activity relationships [46]. Additionally, computational modeling and machine learning tools [47] could assist in predicting effective structural motifs for enhanced efficacy.

### 4.2. Poor Stability and Environmental Durability

Urushiol’s oxidation-prone phenolic groups can degrade under harsh conditions (e.g., UV light and humidity), compromising long-term stability. Therefore, it is essential to develop stimuli-responsive systems (e.g., light- or pH-triggered release mechanisms) and bioinspired self-healing coatings using reversible chemical bonds, which can repair microcracks and maintain barrier function under stress. Systematic durability assessments under extreme conditions (temperature, salinity, and wear) should also be standardized.

### 4.3. Limited Functional Versatility

Current urushiol-based coatings often focus on antibacterial performance alone, limiting their broader applicability. Future directions include synergistic integration with nanomaterials (e.g., AgNPs, TiO_2_, and nepheline) [48,49,50,51] or natural antimicrobial agents (e.g., peptides and essential oils) [52,53,54,55] to achieve multi-target activity. For instance, urushiol–peptide hybrids could disrupt both membrane integrity and intracellular metabolism, providing a solution to antibiotic-resistant strains.

### 4.4. Biocompatibility and Green Manufacturing

For biomedical translation, biocompatibility and toxicity remain key concerns. Although urushiol is natural, residual allergens or byproducts require comprehensive safety assessments. Green synthesis techniques and scalable, eco-friendly production processes must also be developed to reduce cost and environmental impact [56,57,58]. Establishing life cycle toxicity databases and ecological risk profiles will support regulatory approval and sustainability.

### 4.5. Application Expansion and Standardization

Beyond planktonic bacteria, urushiol’s potential against biofilms and enveloped viruses is underexplored. Surface micro/nanostructure engineering (e.g., nanoneedle arrays) [59,60] combined with urushiol’s lipid-disrupting activity could yield dual-function surfaces with both anti-adhesive and virucidal capabilities. Adopting standardized testing protocols (e.g., ISO 22196 [61] and JIS Z 2801 [62]) will enable reliable performance benchmarking and commercialization [63].

### 4.6. Interdisciplinary Collaboration

Addressing the “activity–stability–safety” trilemma demands interdisciplinary collaboration across materials science, microbiology, toxicology, and engineering. Through coordinated efforts, urushiol-based antimicrobial coatings may evolve into sustainable, high-performance solutions for healthcare, marine, and environmental applications.

## 5. Conclusions

Urushiol-based antimicrobial coatings demonstrate remarkable potential in next-generation antimicrobial materials due to their unique molecular architecture and multi-mechanistic bactericidal action. As the core bioactive component of natural lacquer, urushiol achieves potent microbial eradication through synergistic interactions between redox-active phenolic hydroxyl groups and membrane-penetrating alkyl chains. Its antimicrobial efficacy stems from dual pathways: (1) hydrogen bonding-mediated enzyme inhibition and membrane disruption, and (2) oxidative stress induction via intracellular reactive oxygen species (ROS) burst. Compared to conventional synthetic agents (e.g., silver ions and quaternary ammonium salts), urushiol-based systems exhibit superior broad-spectrum activity and environmental compatibility, as well as a reduced risk of resistance development.

Recent advancements in molecular engineering, nanocomposite integration, and dynamic self-healing designs have significantly enhanced the environmental adaptability and long-term stability of urushiol coatings. The development of metal–urushiol coordination complexes and organic–inorganic hybrid systems enables precise control over antimicrobial performance and multifunctional integration. Nevertheless, reconciling the trilemma of “high activity-durability-biosafety” remains a critical challenge for large-scale applications. Future endeavors should prioritize multi-scale structural optimization, stimuli-responsive delivery systems, and comprehensive lifecycle assessment protocols. Concurrently, expanding research into antibiofilm and antiviral functionalities will unlock new applications in healthcare and public hygiene. Through the interdisciplinary convergence of materials science, synthetic biology, and clinical medicine, urushiol-based coatings are poised to revolutionize microbial control strategies in medical devices, marine antifouling, and food packaging, thereby offering sustainable solutions to global microbial contamination challenges.

## Figures and Tables

**Figure 1 polymers-17-01500-f001:**
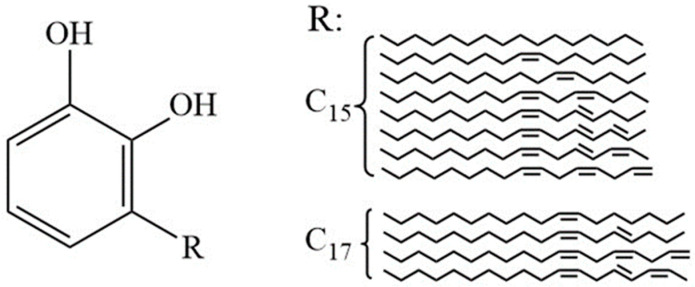
Structure and content of urushiol in Chinese lacquer.

**Figure 2 polymers-17-01500-f002:**
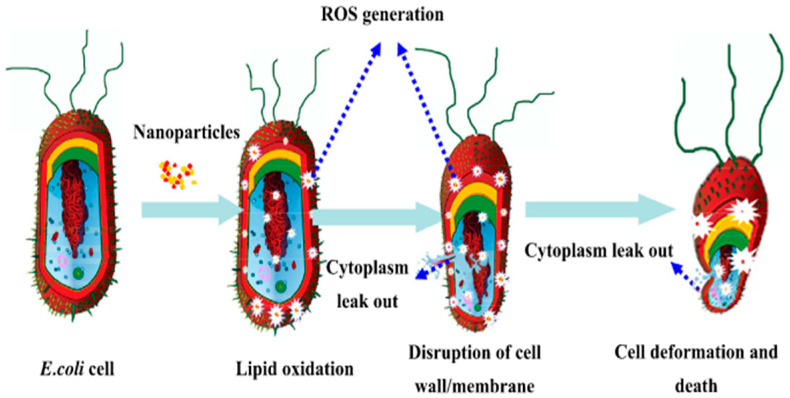
Schematic diagram of the synergistic antibacterial mechanism of copper/TiO_2_ nanoparticles [27].

**Figure 3 polymers-17-01500-f003:**
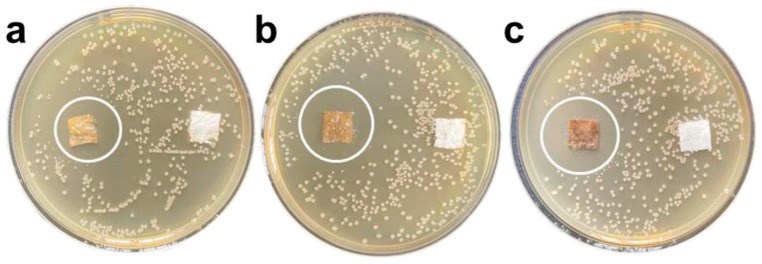
Antibacterial zone diagram of composite antibacterial coating on *E. coli*. The mass fractions of GZC antibacterial powder are 1% (**a**), 3% (**b**), and 5% (**c**) [28].

**Figure 4 polymers-17-01500-f004:**
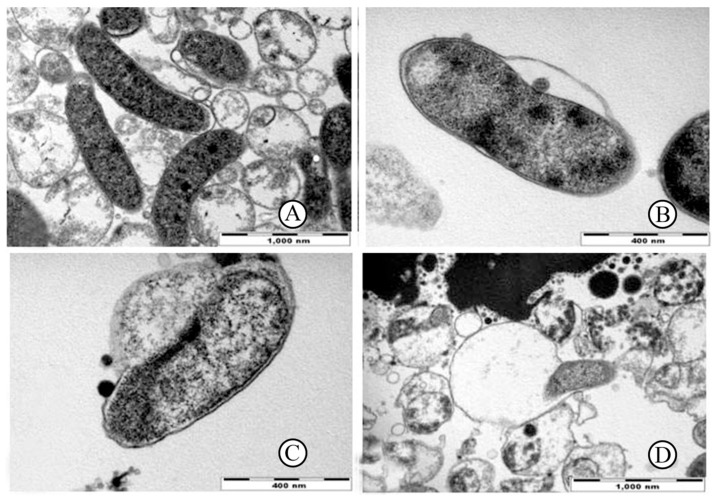
Serial transmission electron micrographs of *H. pylori* exposed to 1×minimal inhibitory concentration of urushiol. (**A**) In control with out urushiol exposure, *H. pylori* bacteria are in normal bacillary form. (**B**) After 3 min exposure, separation of the cell wall and vacuole and bleb formation are observed (arrow). Leakage of some cellular material from the cytoplasmic membrane is observed (arrow head). (**C**) At 6 min after exposure, significant separation with secretory granule loss and lysis of the cytoplasmic membrane are observed (arrow). (**D**) After 10 min exposure, the bacteria are almost complete lysed (arrow) [38].

**Figure 5 polymers-17-01500-f005:**
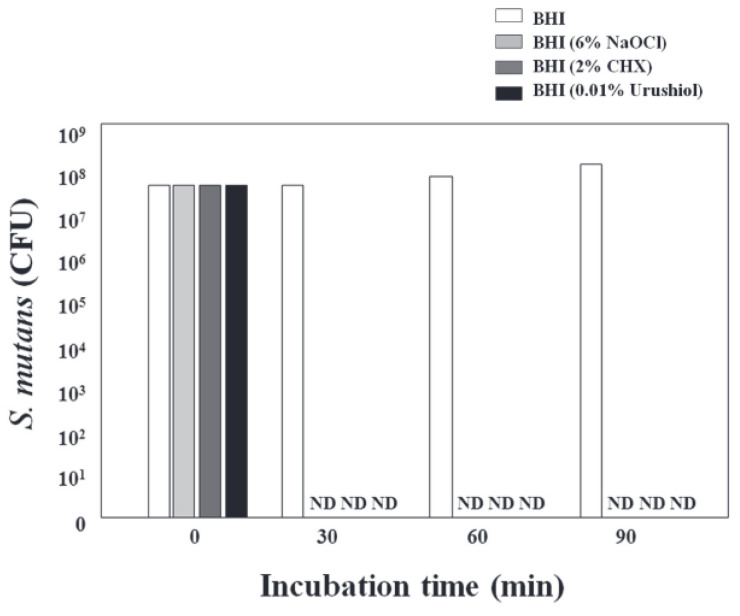
Time-dependent bactericidal activity of urushiol coatings compared to CHX and NaOCl. CFU quantification after 30 min of exposure. ND: not detected [17].

**Table 1 polymers-17-01500-t001:** Antibacterial rate of GZC antibacterial powder against *Escherichia coli* and *Staphylococcus aureus* [28].

	Sample Serial Number	Clamp Count/Individual	Antibacterial Ratio/%
*E. coli*	GZC-1	0	100
	GZC-2	0	100
	GZC-3	0	100
	Control group	630	/
*S. aureus*	GZC-1	0	100
	GZC-2	0	100
	GZC-3	0	100
	Control group	210	/

**Table 2 polymers-17-01500-t002:** Inhibition rates of UL against *Staphylococcus aureus* and *Escherichia coli* [43].

Sample ID	A	B	C	D
*Staphylococcus aureus* inhibition rate/%	0	97	98.7	100
*Escherichia coli* inhibition rate/%	0	80	90	100

## Data Availability

Data are contained within the article.

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
