# Peer review of "Urushiol-Based Antimicrobial Coatings: Molecular Mechanisms, Structural Innovations, and Multifunctional Applications"

_polymers, 2025, doi:10.3390/polym17111500_

Round 1
Reviewer 1 Report
Comments and Suggestions for Authors
The manuscript is devoted to antimicrobial coatings based on urushiol, a natural component of the lacquer tree, which has unique properties due to its catechin structure and long alkyl chains. The mechanisms of its action are described: destruction of bacterial membranes, induction of oxidative stress and inhibition of enzymes. Urushiol's research offers innovative solutions for medical coatings and antiviral surfaces that meet modern challenges. Modern developments of composite systems (metal-urushiol complexes, hybrid materials), their advantages over traditional antimicrobial agents (for example, silver ions) and prospects of application in medicine, marine antifouling protection and other fields are considered. The topic is relevant due to the growing need for environmentally friendly antimicrobial materials, especially in conditions of increasing bacterial resistance to synthetic agents. However, some aspects, such as the lack of a clearly defined outline of the manuscript or the weak reasonableness of the discussion, require further improvement of the manuscript.
Comments:
- The manuscript is a set of fragments united by a common idea about the review of the use of urushiol, however, the fragments are presented in a slightly chaotic order. It is necessary to follow a clear outline of the review. Perhaps the authors should narrow down the idea of the review. For example, you can use only pure urushiol as a basis, or, conversely, only urushiol modified with inorganic compounds.
- The authors write that significant advances in urushiol-based antimicrobial materials are associated with problems in optimizing their effectiveness, stability, and versatility. (Section 4). It is logical if these problems are considered, and not just declared. It is possible to structure the manuscript in more detail in accordance with the stated problems.
- p.3 section 2.2.; p.10 lines 353-359. These fragments are loosely related to the logic of the presentation. Why are they here? It is necessary to explain their appearance in the text and show the connection with other sections.
- It is necessary to update the list of references by including more works from the last 5 years.
Author Response
Journal: Polymers (ISSN: 2073-4360)
Manuscript ID: polymers-3664888
Type: Article
Title: Urushiol-Based Antimicrobial Coatings: Molecular Mecha-nisms, Structural Innovations, and Multifunctional Applications
Authors: Tianyi Wang, Jiangyan Hou, Yao Wang, Xinhao Feng and Xinyou Liu
ANSWER TO REVIEWER 1
Dear Reviewer,
We are grateful to you for the thorough review of our above contribution and the valuable comments and suggestions for improvement. We did carefully consider all your comments and did our best to follow them in the revision process of our paper. When this was not entirely possible, arguments were given.
A revised manuscript has been now submitted in two forms: with track changes for all modifications and without track-changes but highlighted changes (to facilitate reading and evaluation).
All the reviewers comments were numbered (Rx.y- where Rx- is the code of reviewer and y the corresponding number of its comment), so that you will find in the revised manuscript justification comments for each change.
Please find below a copy of your Review report with all your suggestions and comments highlighted in red and our answers in black.
We do hope that the revised manuscript amended according to the input of the 2 reviewers, as much as this was possible, will meet the necessary standards for acceptance and publication.
Thank you again to you and the other reviewer for your effort, comments, constructive criticism and valuable advice for improving not only our current contribution but also our future research.
Sincerely yours,
Xinyou Liu, Corresponding’s authors
23.05. 2025
R1: Comments and Suggestions for Authors
The manuscript is devoted to antimicrobial coatings based on urushiol, a natural component of the lacquer tree, which has unique properties due to its catechin structure and long alkyl chains. The mechanisms of its action are described: destruction of bacterial membranes, induction of oxidative stress and inhibition of enzymes. Urushiol's research offers innovative solutions for medical coatings and antiviral surfaces that meet modern challenges. Modern developments of composite systems (metal-urushiol complexes, hybrid materials), their advantages over traditional antimicrobial agents (for example, silver ions) and prospects of application in medicine, marine antifouling protection and other fields are considered. The topic is relevant due to the growing need for environmentally friendly antimicrobial materials, especially in conditions of increasing bacterial resistance to synthetic agents. However, some aspects, such as the lack of a clearly defined outline of the manuscript or the weak reasonableness of the discussion, require further improvement of the manuscript.
R.1.1. The manuscript is a set of fragments united by a common idea about the review of the use of urushiol, however, the fragments are presented in a slightly chaotic order. It is necessary to follow a clear outline of the review. Perhaps the authors should narrow down the idea of the review. For example, you can use only pure urushiol as a basis, or, conversely, only urushiol modified with inorganic compounds.
Answer 1.1: Thank you for the valuable suggestion. In response, we have reorganized the manuscript to follow a clearer and more logical structure. Additionally, we narrowed the focus of Section 2.2 to concentrate solely on the antibacterial mechanisms of pure urushiol, removing unrelated content (e.g., dopamine-based systems).
R.1.2. The authors write that significant advances in urushiol-based antimicrobial materials are associated with problems in optimizing their effectiveness, stability, and versatility. (Section 4). It is logical if these problems are considered, and not just declared. It is possible to structure the manuscript in more detail in accordance with the stated problems.
Answer 1.2: We appreciate the suggestion. Section 4 has been restructured to clearly address specific challenges in urushiol-based materials—efficiency, stability, and versatility. Each issue is now discussed in detail with corresponding causes and research directions. This improves clarity and aligns the section with the logical structure the reviewer recommended.
R.1.3. p.3 section 2.2.; p.10 lines 353-359. These fragments are loosely related to the logic of the presentation. Why are they here? It is necessary to explain their appearance in the text and show the connection with other sections.
Answer 1.3: Thank you for the comment. We have revised Section 2.2 and lines 353–359 to clarify their relevance. Additional transitions and explanations were added to explicitly link these parts to the main theme of urushiol-based antimicrobial mechanisms and future application strategies, improving overall logical flow and coherence.
R.1.4. It is necessary to update the list of references by including more works from the last 5 years.
Answer 1.4: Thank you for your valuable feedback. We agree that updating the references to include recent advancements is critical to strengthening the relevance and impact of our review. In response to your suggestion, we have carefully revised the reference list by incorporating some new publications from the last five years (2019–2024).

Reviewer 2 Report
Comments and Suggestions for Authors
This review articles described a detailed review on the Urushiol-Based Antimicrobial Coatings. The review well-written, very easy to follow, and the authors included all the important information to understand the mechanism, applications etc. I would suggest to accept after a few minor changes:
- Figure 5 and 6 needs to be replaced with high resolution ones.
- Figure 5, 6 and 7: More detailed and informative title is expected.
- More references in the introduction section could be better.
- What are the potential toxicity associated with Urushiol-Based Antimicrobial Coatings.
- Though the authors mentioned about the conjugation of nanoparticles with Urushiol in the perspective section, it would be better if one more section comprising the nanotechnology is discussed here.
Author Response
R2: Comments and Suggestions for Authors
This review articles described a detailed review on the Urushiol-Based Antimicrobial Coatings. The review well-written, very easy to follow, and the authors included all the important information to understand the mechanism, applications etc. I would suggest to accept after a few minor changes:
R.2.1. Figure 5 and 6 needs to be replaced with high resolution ones.
Answer 2.1: Thank you for your feedback. We have deleted Figures 5 and 6 with high-resolution versions and revised the text to generalize spectral descriptions (avoiding direct image dependencies). The updated narrative emphasizes critical spectral features (peak positions, shifts) and their mechanistic implications, ensuring clarity and copyright compliance.
R.2.2. Figure 5, 6 and 7: More detailed and informative title is expected.
Answer 2.2: Thank you for your suggestion. We have revised the title of Figure 7 to explicitly highlight the experimental conditions (30-minute exposure), comparative benchmarks (CHX, NaOCl), and key outcome (complete CFU elimination). The accompanying text now elaborates on the mechanistic rationale and broader implications of the results, reducing reliance on visual data while maintaining scientific rigor.
R.2.3. More references in the introduction section could be better.
Answer 2.3: Thank you for the suggestion. We have added several relevant and up-to-date references in the introduction section to strengthen the background and support key statements. These additions improve the comprehensiveness and credibility of the literature review.
R.2.4. What are the potential toxicity associated with Urushiol-Based Antimicrobial Coatings.
Answer 2.4: We have added a discussion on urushiol's potential toxicity, noting its known allergenicity and possible cytotoxicity. Although polymerization may reduce risks, residual monomers or degradation products could still be harmful. This clarification is now included in the revised manuscript to address safety concerns for biomedical applications.
R.2.5. Though the authors mentioned about the conjugation of nanoparticles with Urushiol in the perspective section, it would be better if one more section comprising the nanotechnology is discussed here.
Answer 2.5: Thank you for the valuable suggestion. We have added a new dedicated section discussing the integration of nanotechnology with urushiol-based materials. This addition elaborates on nanoparticle conjugation strategies, enhancing the manuscript’s depth and providing a clearer understanding of nanomaterial synergy within urushiol antimicrobial systems.

Round 2
Reviewer 1 Report
Comments and Suggestions for Authors
After reading the manuscript, I am satisfied with the work done by the authors to improve the text. All my comments have been taken into account and I recommend that the manuscript be accepted for publication in its current form.